# Obtaining Spatial Variations in Cabernet Sauvignon (*Vitis vinifera* L.) Wine Flavonoid Composition and Aromatic Profiles by Studying Long-Term Plant Water Status in Hyper-Arid Seasons

Runze Yu [1],*, Nazareth Torres [2] and Sahap Kaan Kurtural [3]

[1] Department of Viticulture and Enology, California State University-Fresno, Fresno, CA 93740, USA
[2] Departamento de Agronomía, Biotecnología y Alimentación, Universidad Pública de Navarra, 31006 Pamplona, Navarra, Spain; nazareth.torres@unavarra.es
[3] Kurtural Vineyard Consulting, Davis, CA 95618, USA; skkurtural@kurturalconsulting.com
* Correspondence: crzyu@csufresno.edu; Tel.: +1-559-278-7112; Fax: +1-559-278-4795

**Abstract:** The spatial variability in vineyard soil might negatively affect wine composition, leading to inhomogeneous flavonoid composition and aromatic profiles. In this study, we investigated the spatial variability in wine chemical composition in a Cabernet Sauvignon (*Vitis vinifera* L.) vineyard in 2019 and 2020. Because of the tight relationships with soil profiles, mid-day stem water potential integrals ($\Psi_{stem}$ Int) were used to delineate the vineyard into two zones, including Zone 1 with relatively higher water stress and Zone 2 with relatively lower water stress. Wine from Zone 2 generally had more anthocyanins in 2019. In 2020, Zone 1 had more anthocyanins and flavonols. Zone 2 had more proanthocyanidin extension and terminal subunits as well as total proanthocyanidins in 2020. According to the Principal Component Analyses (PCA) for berry and wine chemical composition, the two zones were significantly different in the studied wine aromatic compounds. In conclusion, this study provides evidence of the possibility of managing the spatial variability of both wine flavonoid composition and aromatic profiles through connecting vineyard soil variability to grapevine season-long water status.

**Keywords:** wine; flavonoids; plant water status; aromatic compounds; spatial variability; precision viticulture

## 1. Introduction

Grape berry secondary metabolites, including flavonoids and aromatic compounds, directly and significantly determine wine sensory characterization and aging potential [1,2]. Flavonoids mainly include three classes, including anthocyanins, flavonols, and proanthocyanidins. Anthocyanins are responsible for wine color, and the composition in their derivatives and hydroxylation also dictates color appearance and stability [3]. Flavonol accumulation in grape skins is tightly connected with solar radiation, and flavonols also play a critical role in co-pigmentation during wine aging [4,5]. Proanthocyanidins constitute flavan-3-ol monomers, and their polymerization and compositional variations can affect the taste and mouthfeel of wine [6]. To manage flavonoid concentration and composition, water deficits, achieved by varying the applied water amount through irrigation in wine grape vineyards, are often purposely used to improve wine color and stability, and other physical and chemical characteristics that contribute to the final wine quality [7,8]. However, when the other abiotic stresses derived from conditions such as heatwaves and excessive solar radiation are too pronounced, a water deficit might lead to an over-stressed situation in grapevines, resulting in flavonoid degradation and instability in grape berries [9].

Besides flavonoids, aromatic compounds play a significant role in determining wine characteristics as well. There are various classes that can contribute to wine flavor, which

are primarily derived directly from grape tissues, or indirectly from the process of fermentation and aging in toasted oak barrels [10]. For the aromatic compounds directly derived from grape tissues, besides cultivar, which is the most crucial contributor, environmental conditions can rule their composition and concentration [11]. Among these environmental factors, water deficits are often utilized to achieve certain flavor intensities or profiles in vineyards [12,13]. It has been shown that water deficits can increase unsaturated fatty acids in Cabernet Sauvignon berries as well as alcohols and esters in wines [14]. Additionally, terpenes and norisoprenoids are often enhanced by water deficits [13,15].

Normally, a vineyard is often managed uniformly, without considering the spatial variability, which eventually causes spatial variations in grapevine berry chemical development and potentially jeopardizes the overall vineyard productivity and berry quality prior to fermentation [16–18]. Recent research provided evidence of spatial variability in either vineyard characteristics (i.e., soils) or the resultant grapevine physiological parameters, which makes the ability to monitor these spatial variabilities and diminish their negative effects on the resultant wines attainable for growers [19,20]. Meanwhile, some studies reported that long-term plant water status could be capable of delineating vineyard parcels into different management zones, which can simultaneously reveal the spatial variability in berry chemistry due to grapevine water status being directly connected with soil profiles as well as grapevine physiology and berry chemistry [18,21]. However, there is a lack of evidence on whether spatial season-long plant water status can capture the spatial variability in grape berry and wine flavonoid composition and wine aromatic profiles.

Overall, this study was conducted to evaluate the feasibility of using long-term water status in space to capture the spatial variability in wine flavonoid composition and aromatic profile. Based on previously established relationships between season-long water status and soil conditions, where the latter factor can be inherently constant, it was hypothesized that vineyard zone delineation based on season-long plant water status can provide a method to obtain spatial variability in grapevine berry and wine chemistry over seasons. Further, such an approach can provide more evidence and knowledge to growers on managing vineyards selectively to minimize detrimental consequences from spatial heterogeneity in berry/wine chemistry.

## 2. Materials and Methods

### 2.1. Experiment Site, Plant Materials, and Weather Information

The experiment site was located in Oakville, Napa County, USA (38.428823, −122.407906), and planted with Cabernet Sauvignon grafted on 3309C (*V. riparia* × *V. rupestris*). The grapevines were planted in 2015 at 1.5 m × 2.0 m (vine × row), and trained to a bilateral cordon on a single-high-wire trellis system. The grapevines were mechanically pruned to a spur height of 100 mm, and 30% of the shoots were removed at E-L stage 17 mechanically to meet production demands, with no further canopy management in either 2019 or 2020.

Weather information at the experiment site was obtained from the California Irrigation Management Information System (CIMIS) station #77, in Oakville, CA, which was approximately 200 m away from the research site. Monthly rain and reference evapotranspiration data ($ET_o$) were acquired to direct irrigation scheduling during the growing season. At the experiment site, irrigation was applied with a drip irrigation system with two 2 L·h$^{-1}$ emitters at each plant, and the schedule was initiated at fruit-set and continued to harvest (20 September 2019 and 22 September 2020) to replace 50% of the $ET_c$ demand. The weekly applied irrigation amounts were calculated as the product of the calculated crop coefficient ($K_c$) and reference evapotranspiration ($ET_o$). $K_c$ was assessed as reported by Williams and Ayars [22]. To calculate the growing degree days (GDD), the maximum and minimum average air temperature was acquired from the CIMIS station, and the average daily air temperature was assessed to calculate GDD, according to

$$GDD = \sum_{1Apr}^{31Oct} \left[ \frac{(T_{Max\ Air} + T_{Min\ Air})}{2} \right] - T_{Base} \tag{1}$$

where $T_{Max\ Air}$ is the maximum average air temperature, $T_{Min\ Air}$ is the minimum average air temperature, and $T_{Base}$ is the base temperature of 10 °C.

### 2.2. Experimental Design and Vineyard Delineation

At the experiment site, we utilized a 30 m × 30 m grid to collect on-site measurements, which contained 14 experimental units with 3 individual adjacent grapevines in each. Geolocations of the center vine in each experimental unit were recorded with a GPS unit (Yuma 2, Trimble Inc., Sunnyvale, CA, USA) connected to a Trimble Pro 6T DGNSS receiver (Trimble Inc., Sunnyvale, CA, USA) to assist with the geospatial analysis.

To assess the plant water status, mid-day stem water potential ($\Psi_{stem}$) measurements were taken biweekly. The measurement procedures were described previously [23]. To indicate the cumulative grapevine water status information, $\Psi_{stem}$ integrals ($\Psi_{stem}$ Int) were calculated by using natural cubic splines [24]. The sum of the values was divided by the number of days between the first and the last measurements in each year to make the data comparable to each individual measurement. Then, the vineyard was delineated into two zones based on the $\Psi_{stem}$ integral spatial interpolation in 2018, including a more-water-stressed Zone 1 and a less-water-stressed Zone 2. Due to the established correlations between $\Psi_{stem}$ integral and soil profiles previously at this site [23], these two management zones were utilized in 2019 and 2020 to guide differential harvesting and fermentation.

### 2.3. Berry Sampling and Berry Primary Metabolite Assessment at Harvest

The sampling procedure was previously reported [23]. Briefly, a total of 95 berries were collected from each experimental unit at harvest in both years. Within the 95 berries, one subset of 55 berries was used for berry must total soluble solids (TSS), pH, titratable acidity (TA), and berry weight assessments. The second subset of 20 berries was for assessing dry berry skin weight and skin flavonoid contents. The third subset was used for 3-isobutyle-2-methoxypyrazine (IBMP) quantification, with 20 berries taken at harvest in both seasons. Must TSS was measured using a digital refractometer (expressed as °Brix, Atago PR-32, Bellevue, WA, USA), while pH and TA were measured with an automated titrator (expressed as g of tartaric acid·L$^{-1}$ of grape must, 862 Compact TitroSampler, Metrohm, Switzerland).

### 2.4. Extraction of Berry Skin Flavonoids at Harvest

The 20 berry skins in the second subset were manually separated from pulp and seeds, and then lyophilized using a freeze-drier (Triad Freeze-Dry System, Labconco, Kansas City, MO, USA). The skins were powdered with a mixing mill (MM400, Retsch, Mammelzen, Germany). To initiate the extraction, 50 mg of dry skin powder was weighed and mixed with 1 mL of methanol/water/7 M hydrochloric acid solution at 4 °C overnight, where, to make 100 mL of extraction solution, 70 mL of methanol was mixed with 29.42 mL of type 1 water and acidified with 0.58 mL 12.1 N hydrochloric acid. The extracts were then centrifuged at 5000 rpm for 10 min, and the supernatants were collected and filtered through PTFE membrane filters (diameter: 13 mm, pore size: 0.45 μm, VWR, Seattle, WA, USA) and finally transferred into HPLC vials before injection.

### 2.5. Winemaking Procedures

Vinification was conducted in 2019 and 2020 at the UC Davis Teaching and Research Winery. The winemaking procedures were performed according to our previous study [18]. In brief, the grapes were harvested on the same days in both seasons, when the berries in Zone 1 reached a TSS of 26.18 °Brix, 3.44 pH, 8.13 g·L$^{-1}$ TA in 2019 and 23.68 °Brix, 3.50 pH, 7.02 g·L$^{-1}$ TA in 2020, and the berries in Zone 2 reached a TSS of 27.08 °Brix, 3.46 pH, 8.80 g·L$^{-1}$ TA in 2019 and 23.29 °Brix, 3.47 pH, 7.45 g·L$^{-1}$ TA in 2020. The grapes were destemmed and crushed once transported into the winery, and the lot from each management zone was divided into 3 replicate fermentation vessels. Then, 50 mg·L$^{-1}$ of SO$_2$ was added to each vessel to prevent oxidation and soaked for 24 h before yeast

inoculation. The yeast inoculation was performed with EC-1118 yeast (Lallemand Lalvin®, Montreal, QC, Canada) to initiate the alcoholic fermentation at 25 °C in each jacketed stainless-steel fermenter controlled by an Integrated Fermentation Control System (T.J fermenters, Cypress Semiconductor Co., San Jose, CA, USA). To keep the skins submerged during fermentation, 2 volumes of juice were pumped over twice per day automatically by the system, except the pump-over/punch-down, which was performed manually in 2019. The alcoholic fermentation was carried out until the residual sugar contents dropped below 3 g·L$^{-1}$. Then, malolactic fermentation was initiated with the addition of Viniflora® *Oenococcus oeni* (Chr. Hansen A/S, Hørsholm, Denmark) at 12 °C in each fermenter and 60% air humidity. The wines were set to attain a free SO$_2$ level of 30 mg·L$^{-1}$ after the completion of malolactic fermentation and then bottled. Wine samples were filtered through PTFE membrane filters (diameter: 13 mm, pore size: 0.45 μm, VWR, Seattle, WA, USA) and transferred directly into HPLC vials for flavonoid analysis.

*2.6. Berry Skin and Wine Chemical Composition Assessment*

The procedure for berry skin and wine flavonoid analysis was described previously by Martínez-Lüscher et al. (2019) [25]. The filtered supernatant of both the skin extracts and wine samples was transferred into HPLC vials before injection. The low-molecular-weight flavonoids in the berry skins and wine were analyzed using a reversed-phase HPLC (Agilent model 1260, Agilent Technologies, Santa Clara, CA, USA) consisting of a vacuum degasser, an autosampler, a quaternary pump, and a diode array detector with a column heater. A C18 reversed-phase HPLC column (LiChrosphere 100 RP-18, 4 × 520 mm$^2$, 5 μm particle size, Agilent Technologies, Santa Clara, CA, USA) was used for this method. Mobile phases and their flow gradient were described previously as well [25].

Proanthocyanidin subunits were analyzed using an acid catalyst in the presence of phloroglucinol, as previously described [26]. Proanthocyanidins were isolated from the flavonoid extracts by using DSC-18 solid-phase extraction (SPE) tubes (bed weight: 1 g, volume: 6 mL, Agilent Technologies, Santa Clara, CA, USA). In brief, the SPE tubes were pre-conditioned with 3 column volumes (18 mL) of methanol followed by 3 column volumes (18 mL) of water. Then, 1 mL of skin flavonoid extracts or wine samples was passed through the column, followed by 3 column volumes (18 mL) of water to remove glycosides and other low-molecular-weight flavonoids. Finally, 3 mL of methanol was applied 3 times (9 mL in total) to elude the remaining proanthocyanidin purity into a 15 mL Falcon tube (ThermoFisher Scientific, Waltham, WA, USA). The methanol in the eluent was evaporated in a freeze-dryer (Cold Trap 7385020; Labconco, Kansas City, MO, USA), and the remaining proanthocyanidin powder was redissolved in 1 mL of methanol. An equal amount of methanolic extracts (0.5 mL) was mixed with a double-strength phloroglucinolysis reagent (0.5 mL, 100 g/L phloroglucinol and 20 g/L ascorbic acid with 0.2 N HCl in methanol) and water, bathed at 50 °C for 20 min to initiate the reaction. Then, the reaction was stopped by adding 200 μL of the sample mixtures with 1 mL of stopping reagent (40 mM aqueous sodium acetate) and analyzed using a reversed-phase HPLC (Agilent model 1260, Agilent Technologies, Santa Clara, CA, USA). The column used for phloroglucinolysis consisted of two Chromolith RP-18e (100 × 4.6 mm) columns (EMD Millipore Corporation, Burlington, MA, USA) serially connected and protected by a guard column with the same material (4 × 4 mm) (EM Science, Gibbstown, NJ, USA). Mobile phases and their flow gradient were described previously [8]. A computer workstation with Agilent OpenLAB (Chemstation edition, version A.02.10) was used for chromatographic analysis.

At harvest in both 2019 and 2020, the IBMP in berries was assessed through a stable isotope dilution assay (SIDA) using a headspace solid-phase microextraction, coupled to a gas chromatograph and a mass spectrometer (HS-SPME-GS-MS) modified from previously developed methods [27,28]. The samples were analyzed with a 6890N GC (Agilent Technologies, Santa Clara, CA, USA), equipped with a split/splitless injector coupled to a 5973 mass selective detector (MSD). A Gerstel MPS2 autosampler (Gerstel Inc. Columbia, MS, USA) and an HP-5ms capillary column (30 m × 0.25 mm with 0.25 μm film thickness,

Agilent Technologies, Santa Clara, CA, USA) were used for HS sampling. MassHunter Qualitative Analysis software (Version B.07.00, Agilent Technologies, Santa Clara, CA, USA) was used to tentatively quantify the volatile compounds. The analytical method for wine volatile compounds was previously reported [29].

### 2.7. Chemicals

All chromatographic solvents were of high-performance liquid chromatography (HPLC)-grade, including the acetonitrile, methanol, hydrochloric acid, and formic acid. These solvents were purchased from Thermo-Fisher Scientific (Santa Clara, CA, USA). Malvidin 3-*O*-glucoside used for anthocyanin identification was purchased from Extrasynthse (Genay, France). Myricetin-3-*O*-glucuronide, myricetin 3-*O*-glucoside, quercetin 3-*O*-glucunoride, quercetin 3-*O*-galactoside, quercetin 3-*O*-glucoside, kaempferol 3-*O*-glucoside, isorhamnetin 3-*O*-glucoside, and syringetin 3-*O*-glucoside used for flavonol identification were purchased from Sigma-Aldrich (St. Louis, MO, USA). Phloroglucinol was purchased from VWR (Visalia, CA, USA). (-)-Epicatechin used for proanthocyanidin subunit identification was purchased from Extrasynthese (Genay, France).

### 2.8. Statistical Analysis

Kriging for $\Psi_{stem}$ was performed in ArcGIS (version 10.6, Esri, Redlands, CA, USA), and *k*-means clustering was performed in R (RStudio, Inc., Boston, MA, USA) with package NbClust, v3.0 [30]. An ordinary kriging method was used for $\Psi_{stem}$ since there was no trend observed in the vineyard in 2018 (Figure 1). The *k*-means clustering analysis and the practical manageability were both considered, and the vineyard was delineated into two significantly different clusters based on $\Psi_{stem}$ integrals, including a severely water-stressed zone (Zone 1) and a moderately water-stressed zone (Zone 2). Based on this delineation, data from the experimental units were grouped together and grapes were harvested differentially according to the locations of the experimental units within each cluster for statistical analysis and winemaking.

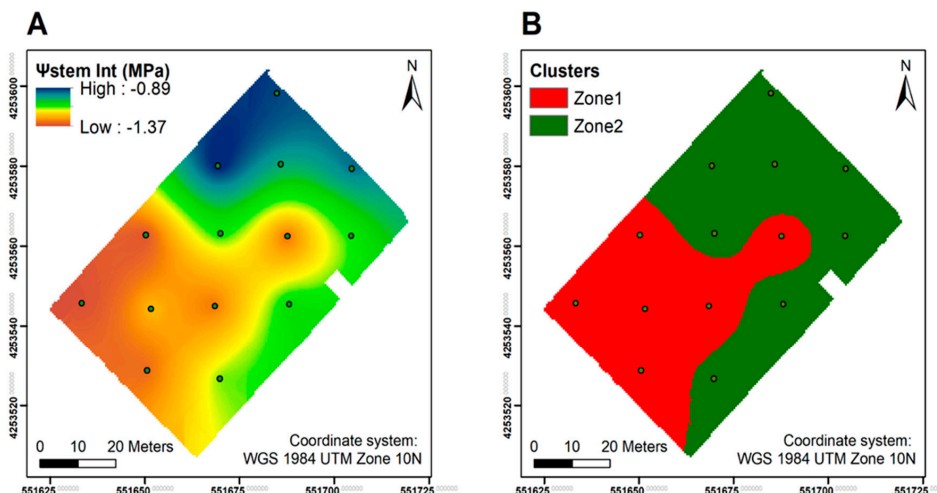

**Figure 1.** Interpolation maps of season-long grapevine water status, presented as stem water potential integrals ($\Psi_{stem}$ Int), and *k*- means clustering maps, delineating the vineyard into two zones to guide the differential harvesting in 2019 and 2020. (**A**) $\Psi_{stem}$ Int kriging map in 2018, (**B**) *k*-means clustering of $\Psi_{stem}$ integrals in 2018. Figure was reanalyzed and adapted from Yu and Kurtural (2020) [23].

All data were tested for normality using the Shapiro–Wilk test, and they were subjected to mean separation by using two-way ANOVA (zoning and year) with the package "stats" in Rstudio (R Foundation for Statistical Computing, Vienna, Austria) [31]. Significant statistical differences were determined when *p* values were less than 0.05. The coefficient of determination between variables was calculated through linear regression analysis; *p* values were acquired to present the significances in the linear fittings. Principle

component analysis (PCA) was performed with the package "stats" to analyze the relationships between yield components and berry/wine chemistry as well as wine aromatic compounds detected in the finished wines. The PCA for individuals and variables was visualized using the package "factoextra" [32].

## 3. Results

### 3.1. Weather at Experiment Site

In the two studied years, the experiment site received drastically different amounts of precipitation, but similar GDD was accumulated (Table 1). There was 919.4 mm of precipitation in 2019 from January to harvest, and almost all the precipitation occurred before the data collection commenced, with only 1.7 mm of precipitation during the growing season from June to September. In 2020, it was hyper-arid, with only 143.6 mm of precipitation at the experiment site, and only 2.3 mm of it was received during the growing season. The two years were similar in GDD accumulation; at the time the fruits were picked, the second year only had 44.8 °C more heat accumulated than the first season. A total of 1859.0 °C and 1903.8 °C of heat was accumulated from 1 April to 31 October in 2019 and 2020, respectively. Regarding daily maximum and minimum air temperature, the second season was hotter than the first season. During the growing season in 2019, there were 18 days and 0 days exceeding a maximum air temperature of 35 °C and 40 °C, respectively. However, in 2020, there were 25 days and 5 days exceeding a maximum air temperature of 35 °C and 40 °C, respectively.

**Table 1.** Weather information at the experiment site as obtained from California Irrigation Management Information System (CIMIS) Station (#77 Oakville, Napa) in 2019 and 2020 [a].

| Year | Month | Precipitation (mm) | GDD (°C) | Air Max Temperature (°C) | Air Min Temperature (°C) |
|------|-------|--------------------|----------|--------------------------|--------------------------|
| 2019 | Jan | 248.5 | - | 15.9 | 4.7 |
| | Feb | 422.2 | - | 12.9 | 2.7 |
| | Mar | 145.6 | 48.5 | 17.5 | 4.9 |
| | Apr | 12.5 | 229.5 | 23.3 | 8.8 |
| | May | 88.9 | 397.0 | 22.4 | 8.4 |
| | Jun | 0.0 | 702.7 | 29.2 | 11.2 |
| | Jul | 0.2 | 1029.2 | 29.9 | 11.1 |
| | Aug | 0.0 | 1393.7 | 31.2 | 12.3 |
| | Sep [b] | 1.5 | 1681.5 | 29.4 | 9.7 |
| | Oct | 0.2 | 1859.0 | 26.6 | 4.9 |
| 2020 | Jan | 58.5 | - | 15.4 | 3.5 |
| | Feb | 1.0 | - | 20.6 | 3.7 |
| | Mar | 29.8 | 31.6 | 17.6 | 4.4 |
| | Apr | 25.9 | 182.1 | 23.0 | 7.1 |
| | May | 26.1 | 414.8 | 26.2 | 8.8 |
| | Jun | 0.2 | 713.6 | 29.5 | 10.4 |
| | Jul | 0.2 | 1027.1 | 30.2 | 10.1 |
| | Aug | 1.6 | 1388.9 | 31.8 | 12.3 |
| | Sep [b] | 0.3 | 1726.3 | 31.4 | 11.1 |
| | Oct | 0.3 | 1903.8 | 29.7 | 8.1 |

[a] Abbreviations: GDD: growing degree days. [b] When the harvests of the data vines took place.

### 3.2. Berry and Wine Chemical Profiles

Between the two zones, there was minimal difference observed in berry skin anthocyanins, except for peonidin-3-acetyl-glucoside in 2019 and $3'4'5'$- to $3'5'$-hydroxylated ratio in 2020, which were higher in Zone 1 (Table 2). Comparing season to season, the second season showed generally more total berry anthocyanins than the first season. Regarding berry skin flavonols, there was no difference between the two zones over the two years (Table 3). Unlike berry skin anthocyanins, the second season generally had lower values of berry skin flavonols than the first season.

**Table 2.** Berry skin anthocyanins at harvest of Cabernet Sauvignon as separated by season-long plant water status in Oakville, CA, USA in 2019 and 2020 [a].

| | 2019 | | | 2020 | | | Year | Zoning | Year × Zoning |
|---|---|---|---|---|---|---|---|---|---|
| | Zone 1 ± SD [b] | Zone 2 ± SD | Pr (>F) | Zone 1 ± SD | Zone 2 ± SD | Pr (>F) | | | |
| **Glucoside** | | | | | | | | | |
| Delphinidin-3-glucoside | 21.61 ± 3.41 | 22.94 ± 3.86 | ns | 21.81 ± 3.31 | 22.83 ± 3.90 | ns | ns | ns | ns |
| Cyanidin-3-glucoside | 1.82 ± 0.45 | 1.88 ± 0.45 | ns | 1.42 ± 0.28 | 1.79 ± 0.52 | ns | ns | ns | ns |
| Petunidin-3-glucoside | 15.11 ± 2.03 | 15.98 ± 2.18 | ns | 16.89 ± 2.43 | 17.03 ± 2.49 | ns | ns | ns | ns |
| Peonidin-3-glucoside | 9.30 ± 1.16 | 9.57 ± 1.35 | ns | 9.09 ± 1.56 | 9.91 ± 1.59 | ns | ns | ns | ns |
| Malvidin-3-glucoside | 91.61 ± 5.97 | 98.73 ± 8.32 | ns | 118.02 ± 12.41 | 113.25 ± 9.65 | ns | <0.001 *** | ns | ns |
| Total glucosides | 139.46 ± 12.34 | 149.11 ± 15.21 | ns | 167.23 ± 19.02 | 164.81 ± 17.36 | ns | 0.002 ** | ns | ns |
| Acetylated | | | | | | | | | |
| Delphinidin-3-acetyl-glucoside | 4.22 ± 0.69 | 4.24 ± 0.66 | ns | 4.52 ± 0.82 | 4.40 ± 0.80 | ns | ns | ns | ns |
| Cyanidin-3-acetyl-glucoside | 0.51 ± 0.14 | 0.45 ± 0.17 | ns | 1.15 ± 0.14 | 1.14 ± 0.11 | ns | <0.001 *** | ns | ns |
| Petunidin-3-acetyl-glucoside | 4.64 ± 0.59 | 4.66 ± 0.54 | ns | 5.27 ± 0.79 | 4.98 ± 0.73 | ns | ns | ns | ns |
| Peonidin-3-acetyl-glucoside | 3.10 ± 0.17 a | 2.84 ± 0.23 b | 0.034 * | 3.10 ± 0.63 | 3.07 ± 0.50 | ns | ns | ns | ns |
| Malvidin-3-acetyl-glucoside | 48.18 ± 3.69 | 48.37 ± 2.99 | ns | 64.55 ± 6.70 | 57.53 ± 6.23 | ns | <0.001 *** | ns | ns |
| Total acetylated | 60.70 ± 4.57 | 60.56 ± 3.67 | ns | 78.58 ± 8.65 | 71.12 ± 7.77 | ns | <0.001 *** | ns | ns |
| Coumarylated | | | | | | | | | |
| Delphinidin-3-*p*-coumaroyl-glucoside | 1.76 ± 0.30 | 1.90 ± 0.29 | ns | 1.97 ± 0.44 | 1.78 ± 0.46 | ns | ns | ns | ns |
| Cyanidin-3-*p*-coumaroyl-glucoside | 0.43 ± 0.06 | 0.44 ± 0.05 | ns | 0.98 ± 0.29 | 0.81 ± 0.20 | ns | <0.001 *** | ns | ns |
| Petunidin-3-*p*-coumaroyl-glucoside | 1.90 ± 0.24 | 1.94 ± 0.27 | ns | 2.16 ± 0.30 | 2.05 ± 0.20 | ns | ns | ns | ns |
| Peonidin-3-*p*-coumaroyl-glucoside | 2.96 ± 0.34 | 2.87 ± 0.19 | ns | 2.82 ± 0.36 | 2.91 ± 0.23 | ns | ns | ns | ns |
| Malvidin-3-*p*-coumaroyl-glucoside | 22.43 ± 1.45 | 23.45 ± 1.68 | ns | 27.99 ± 2.67 | 26.35 ± 1.75 | ns | <0.001 *** | ns | ns |
| Total coumarylated | 29.48 ± 2.05 | 30.59 ± 2.23 | ns | 35.89 ± 2.77 | 33.90 ± 2.47 | ns | <0.001 *** | ns | ns |
| $3'5'$-hydroxylated | 18.13 ± 2.03 | 18.05 ± 2.13 | ns | 18.55 ± 3.07 | 19.63 ± 2.78 | ns | ns | ns | ns |
| $3'4'5'$-hydroxylated | 211.51 ± 15.71 | 222.20 ± 17.82 | ns | 263.16 ± 27.97 | 250.20 ± 23.09 | ns | <0.0001 *** | ns | ns |
| $3'4'5'/3'5'$-hydroxylated ratio | 11.72 ± 0.76 | 12.41 ± 1.28 | ns | 14.30 ± 0.87 a | 12.86 ± 1.18 b | 0.028 * | 0.003 ** | 0.017 * | ns |
| Total anthocyanins | 229.65 ± 17.43 | 240.25 ± 19.03 | ns | 281.70 ± 30.96 | 269.83 ± 25.25 | ns | <0.0001 *** | ns | ns |

[1] n = 13; ns: not significant, different letters indicate significant mean separation according to Tukey's HSD test ($p < 0.05$). [2] Zone 1: severe water stress zone, Zone 2: moderate water stress zone, numbers in the column were expressed in mg per 100 g berry fresh weight and as their means ± one standard deviation. [3] Letters represent ranking after Tukey's *post hoc* analyses. Asterisks represents significant levels *p*, ***: $p < 0.001$; **: $p < 0.01$; *: $p < 0.05$.

**Table 3.** Berry skin flavonols at harvest of Cabernet Sauvignon as separated by season-long plant water status in Oakville, CA, USA in 2019 and 2020 [a].

| | 2019 | | | 2020 | | | Year | Zoning |
|---|---|---|---|---|---|---|---|---|
| | **Zone 1 ± SD** [b] | **Zone 2 ± SD** | **Pr (>F)** | **Zone 1 ± SD** | **Zone 2 ± SD** | **Pr (>F)** | | |
| Myricetin-3-galactoside | 2.95 ± 0.56 | 2.77 ± 0.20 | ns | 2.74 ± 0.34 | 2.71 ± 0.17 | ns | ns | ns |
| Myricetin-3-glucoside | 0.95 ± 0.17 | 0.90 ± 0.07 | ns | 0.79 ± 0.12 | 0.74 ± 0.07 | ns | <0.001 *** | ns |
| Quercetin-3-galactoside | 0.45 ± 0.18 | 0.38 ± 0.08 | ns | 0.48 ± 0.14 | 0.49 ± 0.07 | ns | ns | ns |
| Quercetin-3-glucoside | 2.80 ± 1.01 | 2.41 ± 0.54 | ns | 2.32 ± 0.58 | 2.36 ± 0.38 | ns | ns | ns |
| Laricetin-3-glucoside | 0.68 ± 0.10 | 0.62 ± 0.09 | ns | 0.71 ± 0.12 | 0.66 ± 0.04 | ns | ns | ns |
| Kaempferol-3-glucoside | 0.69 ± 0.07 | 0.65 ± 0.08 | ns | 0.51 ± 0.15 | 0.47 ± 0.11 | ns | <0.001 *** | ns |
| Isorhamnetin-3-glucoside | 1.29 ± 0.09 | 1.30 ± 0.06 | ns | 0.58 ± 0.20 | 0.51 ± 0.07 | ns | <0.001 *** | ns |
| Syringetin-3-glucoside | 0.64 ± 0.04 | 0.67 ± 0.05 | ns | 0.79 ± 0.10 | 0.70 ± 0.06 | ns | <0.001 *** | ns |
| $3'5'$-hydroxylated | 4.54 ± 1.15 | 4.09 ± 0.61 | ns | 3.38 ± 0.90 | 3.36 ± 0.52 | ns | 0.005 ** | ns |
| $3'4'5'$-hydroxylated | 5.22 ± 0.84 | 4.96 ± 0.30 | ns | 5.03 ± 0.66 | 4.81 ± 0.29 | ns | ns | ns |
| $3'4'5'/3'5'$-hydroxylated ratio | 1.18 ± 0.15 | 1.23 ± 0.15 | ns | 1.53 ± 0.22 | 1.47 ± 0.25 | ns | <0.001 *** | ns |
| Total flavonols | 10.45 ± 1.97 | 9.70 ± 0.84 | ns | 9.13 ± 1.60 | 8.83 ± 0.56 | ns | 0.033 * | ns |

[1] n = 13, ns: not significant, different letters indicate significant mean separation according to Tukey's HSD test ($p < 0.05$). [2] Zone 1: severe water stress zone, Zone 2: moderate water stress zone, numbers in the column were expressed in mg per 100 g berry fresh weight and as their means ± one standard deviation. [3] Letters represent ranking after Tukey's *post hoc* analyses. Asterisks represents significant levels *p*, ***: $p < 0.001$; **: $p < 0.01$; *: $p < 0.05$.

In wine, the two zones showed more differences than what was observed in the berries. For wine anthocyanins, Zone 2 had higher concentrations in all the anthocyanin derivatives except delphinidin-3-acetyl-glucoside, cyanidin-3-acetyl-glucoside, peonidin-3-acetyl-glucoside, and cyanidin-3-coumaroyl-glucoside, which did not differ between the two zones (Table 4). Thus, wines from Zone 2 had higher concentration of total glucosides, acetylated and coumaroylated anthocyanins, and total anthocyanins. $3'4'5'$- and $3'5'$-hydroxylated anthocyanin totals were higher in Zone 2, and the ratio between the two parameters was also higher in 2019. However, the opposite results were observed in 2020, where Zone 1 generally had higher concentrations of most of the anthocyanin derivatives, with the exception of cyanidin-3-glucoside, peonidin-3-glucoside, peonidin-3-acetyl-glucoside, cyanidin-3-coumaroyl-glucoside, and the $3'4'5'$-to-$3'5'$-hydroxylated ratio, which did not differ between zones. In addition, the second season had higher anthocyanin concentrations in general compared to the first season.

For wine flavonols, in 2019, Zone 1 had higher concentrations of quercetin-3-glucoside, kaempferol-3-glucoside, and $3'5'$-hydroxylated flavonols (Table 5). Zone 2 had a higher concentration of myricetin-3-galactoside, myricetin-3-glucoside, $3'4'5'$-hydroxylated flavonols, and $3'4'5$-to-$3'5'$-hydroxylated ratio. In 2020, most of the flavonol derivatives were greater in Zone 1 than in Zone 2. Regarding wine proanthocyanidin composition, Zone 2 had higher EC and ECG extension subunits, a higher total proanthocyanidin concentration, and higher mDP (Table 6). In 2020, most of the subunits had higher concentrations in Zone 2, except for ECG extension, EC, and ECG terminal subunits.

**Table 4.** Wine anthocyanin concentration at harvest of Cabernet Sauvignon as separated by season-long plant water status in Oakville, CA, USA in 2019 and 2020 [a].

| | 2019 | | | 2020 | | | Year | Zoning | Year × Zoning |
|---|---|---|---|---|---|---|---|---|---|
| | Zone 1 ± SD [b] | Zone 2 ± SD | Pr (>F) | Zone 1 ± SD | Zone 2 ± SD | Pr (>F) | | | |
| | | | | Glucoside | | | | | |
| Delphinidin-3-glucoside | 7.41 ± 2.46 b | 14.91 ± 2.56 a | <0.001 *** | 21.51 ± 0.87 a | 19.85 ± 0.36 b | 0.001 ** | <0.001 *** | <0.001 *** | <0.001 *** |
| Cyanidin-3-glucoside | 0.62 ± 0.10 b | 0.90 ± 0.11 a | <0.001 *** | 1.35 ± 0.22 | 1.16 ± 0.01 | ns | <0.001 *** | ns | <0.001 *** |
| Petunidin-3-glucoside | 16.54 ± 2.81 b | 8.90 ± 2.91 a | <0.001 *** | 27.99 ± 1.43 a | 24.43 ± 0.21 b | <0.001 *** | <0.001 *** | 0.030 * | <0.001 *** |
| Peonidin-3-glucoside | 4.33 ± 1.31 b | 7.48 ± 0.77 a | <0.001 *** | 15.05 ± 1.36 | 13.41 ± 1.27 | ns | <0.001 *** | ns | <0.001 *** |
| Malvidin-3-glucoside | 107.90 ± 31.80 b | 184.00 ± 23.99 a | <0.001 *** | 421.82 ± 13.45 a | 352.50 ± 1.12 b | <0.001 *** | <0.001 *** | ns | <0.001 *** |
| Total glucosides | 129.16 ± 38.54 b | 223.82 ± 30.17 a | <0.001 *** | 487.73 ± 17.34 a | 411.35 ± 0.72 b | <0.001 *** | <0.001 *** | ns | <0.001 *** |
| | | | | Acetylated | | | | | |
| Delphinidin-3-acetyl-glucoside | 5.81 ± 0.22 | 5.92 ± 0.62 | ns | 7.49 ± 0.42 a | 6.12 ± 0.20 b | <0.001 *** | <0.001 *** | 0.001 ** | <0.001 *** |
| Cyanidin-3-acetyl-glucoside | 1.23 ± 0.41 | 0.93 ± 0.11 | ns | 5.03 ± 0.37 a | 4.05 ± 0.03 b | <0.001 *** | <0.001 *** | <0.001 *** | 0.001 ** |
| Petunidin-3-acetyl-glucoside | 2.90 ± 1.02 b | 5.31 ± 0.75 a | <0.001 *** | 9.52 ± 0.09 a | 7.41 ± 0.08 b | <0.001 *** | <0.001 *** | ns | <0.001 *** |
| Peonidin-3-acetyl-glucoside | 0.68 ± 0.10 | 0.77 ± 0.06 | ns | 2.04 ± 0.46 | 1.57 ± 0.26 | ns | <0.001 *** | ns | 0.022 * |
| Malvidin-3-acetyl-glucoside | 44.64 ± 14.44 b | 76.84 ± 10.57 a | 0.001 ** | 192.79 ± 4.51 a | 166.39 ± 0.93 b | <0.001 *** | <0.001 *** | ns | <0.001 *** |
| Total acetylated | 55.27 ± 15.33 b | 89.77 ± 10.75 a | 0.001 ** | 216.88 ± 4.93 a | 185.54 ± 0.92 b | 0.001 ** | <0.001 *** | ns | <0.001 *** |
| | | | | Coumarylated | | | | | |
| Delphinidin-3-*p*-coumaroyl-glucoside | 2.04 ± 0.63 b | 3.66 ± 0.58 a | 0.001 ** | 6.36 ± 0.43 a | 5.67 ± 0.07 b | 0.002 ** | <0.001 *** | 0.029 ** | <0.001 *** |
| Cyanidin-3-*p*-coumaroyl-glucoside | 1.35 ± 0.16 | 1.54 ± 0.32 | ns | 2.44 ± 0.20 | 2.26 ± 0.09 | ns | <0.001 *** | ns | 0.048 * |
| Petunidin-3-*p*-coumaroyl-glucoside | 0.66 ± 0.21 b | 0.99 ± 0.24 a | 0.030 * | 2.40 ± 0.28 a | 2.10 ± 0.01 b | 0.030 * | <0.001 *** | ns | 0.002 ** |
| Peonidin-3-*p*-coumaroyl-glucoside | 0.47 ± 0.16 b | 0.96 ± 0.20 a | <0.001 *** | 2.37 ± 0.30 | 2.33 ± 0.01 | ns | <0.001 *** | 0.011 * | 0.004 ** |
| Malvidin-3-*p*-coumaroyl-glucoside | 7.64 ± 2.87 b | 16.01 ± 2.69 a | <0.001 *** | 46.13 ± 4.02 a | 42.12 ± 0.59 b | 0.036 * | <0.001 *** | ns | <0.001 *** |
| Total coumarylated | 12.16 ± 3.68 b | 23.14 ± 4.01 a | <0.001 *** | 59.70 ± 5.23 a | 54.49 ± 0.75 b | 0.037 * | <0.001 *** | ns | <0.001 *** |
| Total anthocyanins | 196.59 ± 57.53 b | 336.74 ± 44.92 a | <0.001 *** | 764.31 ± 27.51 a | 651.38 ± 2.38 b | <0.001 *** | <0.001 *** | ns | <0.001 *** |
| 3′5′-hydroxylated | 8.69 ± 1.38 b | 12.57 ± 1.39 a | <0.001 *** | 28.29 ± 2.18 a | 24.78 ± 1.08 b | 0.005 ** | <0.001 *** | ns | <0.001 *** |
| 3′4′5′-hydroxylated | 187.90 ± 56.18 b | 324.17 ± 43.55 a | <0.001 *** | 736.02 ± 25.33 a | 626.60 ± 1.30 b | <0.001 *** | <0.001 *** | ns | <0.001 *** |
| 3′4′5′/3′5′-hydroxylated ratio | 21.23 ± 3.25 b | 25.74 ± 0.79 a | 0.008 ** | 26.09 ± 1.11 | 25.32 ± 1.05 | ns | 0.008 ** | 0.022 * | 0.002 ** |

[1] n = 13, ns: not significant, different letters indicate significant mean separation according to Tukey's HSD test ($p < 0.05$), [2] Zone 1: severe water stress zone, Zone 2: moderate water stress zone, numbers in the column were expressed in mg per L and as their means ± one standard deviation, [3] Letters represent ranking after Tukey's *post hoc* analyses. Asterisks represents significant levels *p*, ***: $p < 0.001$; **: $p < 0.01$; *: $p < 0.05$.

**Table 5.** Wine flavonol concentration at harvest of Cabernet Sauvignon as separated by season-long plant water status in Oakville, CA, USA in 2019 and 2020 [a].

| | 2019 | | | 2020 | | | Year | Zoning | Year × Zoning |
|---|---|---|---|---|---|---|---|---|---|
| | Zone 1 ± SD [b] | Zone 2 ± SD | Pr (>F) | Zone 1 ± SD | Zone 2 ± SD | Pr (>F) | | | |
| Myricetin-3-galactoside | 1.80 ± 0.13 b | 2.07 ± 0.12 a | 0.003 ** | 3.02 ± 0.01 a | 2.20 ± 0.07 b | <0.001 *** | <0.001 *** | <0.001 *** | <0.001 *** |
| Myricetin-3-glucoside | 10.41 ± 0.12 b | 11.77 ± 0.38 a | <0.001 *** | 21.20 ± 0.28 a | 16.16 ± 0.28 b | <0.001 *** | <0.001 *** | <0.001 *** | <0.001 *** |
| Quercetin-3-galactoside | 0.82 ± 0.05 | 0.81 ± 0.05 | ns | 0.80 ± 0.03 a | 0.60 ± 0.03 b | <0.001 *** | <0.001 *** | <0.001 *** | <0.001 *** |
| Quercetin-3-glucoside | 6.42 ± 0.26 a | 4.20 ± 0.06 b | <0.001 *** | 7.31 ± 0.15 a | 6.01 ± 0.42 b | <0.001 *** | <0.001 *** | <0.001 *** | <0.001 *** |
| Laricetin-3-glucoside | 2.99 ± 0.08 | 2.94 ± 0.03 | ns | 3.83 ± 0.10 a | 2.95 ± 0.06 b | <0.001 *** | <0.001 *** | <0.001 *** | <0.001 *** |
| Kaempferol-3-glucoside | 0.50 ± 0.03 a | 0.35 ± 0.11 b | 0.011 * | 0.34 ± 0.03 a | 0.27 ± 0.04 b | 0.004 ** | <0.001 *** | <0.001 *** | ns |
| Isorhamnetin-3-glucoside | 3.48 ± 0.30 | 3.45 ± 0.93 | ns | 3.51 ± 0.44 a | 2.03 ± 0.13 b | <0.001 *** | <0.001 *** | <0.001 *** | <0.001 *** |
| Syringetin-3-glucoside | 4.68 ± 0.13 | 4.59 ± 0.28 | ns | 7.24 ± 0.35 a | 5.94 ± 0.10 b | <0.001 *** | <0.001 *** | <0.001 *** | <0.001 *** |
| 3′5′-hydroxylated | 10.72 ± 0.58 a | 8.46 ± 0.93 b | <0.001 *** | 11.62 ± 0.55 a | 8.64 ± 0.32 b | <0.001 *** | ns | <0.001 *** | ns |
| 3′4′5′-hydroxylated | 19.88 ± 0.25 b | 21.37 ± 0.42 a | <0.001 *** | 35.29 ± 0.73 a | 27.25 ± 0.52 b | <0.001 *** | <0.001 *** | <0.001 *** | <0.001 *** |
| 3′4′5′/3′5′-hydroxylated ratio | 1.86 ± 0.08 b | 2.55 ± 0.27 a | <0.001 *** | 3.05 ± 0.21 | 3.16 ± 0.06 | ns | <0.001 *** | <0.001 *** | <0.001 *** |
| Total flavonols | 31.09 ± 0.82 | 30.18 ± 0.98 | ns | 47.25 ± 0.21 a | 36.15 ± 0.88 b | <0.001 *** | <0.001 *** | <0.001 *** | <0.001 *** |

[1] n = 13, ns: not significant, different letters indicate significant mean separation according to Tukey's HSD test ($p < 0.05$), [2] Zone 1: severe water stress zone, Zone 2: moderate water stress zone, numbers in the column were expressed in mg per L and as their means ± one standard deviation, [3] Letters represent ranking after Tukey's *post hoc* analyses. Asterisks represents significant levels $p$, ***: $p < 0.001$; **: $p < 0.01$; *: $p < 0.05$.

**Table 6.** Wine proanthocyanidin concentration at harvest of Cabernet Sauvignon as separated by season-long plant water status in Oakville, CA, USA in 2019 and 2020 [a].

| | | 2019 | | | 2020 | | | Year | Zoning | Year × Zoning |
|---|---|---|---|---|---|---|---|---|---|---|
| | | Zone 1 ± SD [b] | Zone 2 ± SD | Pr (>F) | Zone 1 ± SD | Zone 2 ± SD | Pr (>F) | | | |
| Extension subunits | EGC [c] | 113.36 ± 59.63 | 352.75 ± 134.59 | ns | 212.44 ± 40.81 b | 281.84 ± 6.16 a | 0.002 ** | ns | 0.014 * | ns |
| | C [c] | 40.78 ± 21.46 | 55.12 ± 7.47 | ns | 66.67 ± 4.63 b | 73.55 ± 5.87 a | 0.048 * | <0.001 *** | ns | ns |
| | EC [c] | 153.43 ± 84.15 b | 501.50 ± 184.02 a | 0.004 *** | 210.72 ± 11.34 b | 285.71 ± 10.37 a | <0.001 *** | ns | 0.017 * | ns |
| | ECG [c] | 9.73 ± 5.48 b | 18.63 ± 6.32 a | 0.033 * | 6.06 ± 1.35 | 6.06 ± 0.44 | ns | 0.002 ** | ns | ns |
| Terminal subunits | C | 61.15 ± 38.43 | 125.73 ± 52.74 | 0.043 * | 97.02 ± 19.54 b | 136.99 ± 15.45 a | 0.003 ** | ns | 0.017 ** | ns |
| | EC | 9.45 ± 5.62 | 9.90 ± 2.64 | ns | 12.79 ± 5.18 | 12.46 ± 2.39 | ns | ns | ns | ns |
| | ECG | 4.56 ± 2.71 | 4.78 ± 1.27 | ns | 6.18 ± 2.50 | 6.02 ± 1.15 | ns | ns | ns | ns |
| Total proanthocyanidins | | 467.87 ± 291.46 b | 1068.42 ± 371.27 a | 0.015 * | 611.88 ± 44.67 b | 802.63 ± 22.04 a | <0.001 *** | ns | 0.017 ** | ns |
| mDP [c] | | 5.27 ± 0.51 b | 7.13 ± 1.73 a | 0.042 * | 6.64 ± 0.98 b | 12.71 ± 2.67 a | <0.001 *** | <0.001 *** | <0.001 *** | 0.003 ** |

[1] n = 13, ns: not significant, different letters indicate significant mean separation according to Tukey's HSD test ($p < 0.05$), [2] Zone 1: severe water stress zone, Zone 2: moderate water stress zone, numbers in the column were expressed in mg per L and as their means ± one standard deviation. [3] Abbreviations: C, (+)-catechin; EC, (−)-epicatechin; ECG, (−)-epicatechin-3-O-gallate; EGC, (−)-epigallocatechin; mDP: mean degree of polymerization. mDP was calculated as the ratio of total proanthocyanidins to the terminal subunits. [4] Letters represent ranking after Tukey's *post hoc* analyses. Asterisks represents significant levels *p*, ***: $p < 0.001$; **: $p < 0.01$; *: $p < 0.05$.

PCA analyses were performed to investigate the relationships among the chemical compounds in grape berries between the two zones (Figure 2), and the two zones were grouped separately in both years according to the main factors. In 2019, the first two principal components (PC) accounted for 77.4% of the total variation in the dataset, and PC1 and PC2 accounted for 45.9% and 31.5% of the total variation, respectively (Figure 2A). Zone 1 was characterized by having a higher 3′4′5′-to-total quercetins ratio, total myricetins, total flavonols, and tri- to di-hydroxylated anthocyanins. Zone 2 was characterized by having higher concentrations of most of the anthocyanin derivatives and total anthocyanins. The zoning effect was significant because the predominant variables in each zone, namely the anthocyanin and flavonol derivatives, were not significantly correlated with each other. In 2020, the first two PC explained 77% of the total variation in the dataset, and PC1 and PC2 accounted for 52.7% and 24.3% of the total variation, respectively (Figure 2B). Zone 1 was characterized by having higher total quercetins, total myricetins, and total flavonols. However, Zone 2 was characterized by having higher concentrations in most of the anthocyanin derivatives and tri- to di-hydroxylated anthocyanins. Similar to 2019, the zoning effect was significant but the variables did not show significant correlations with each other; rather, they were significantly correlated within each chemical family group, including anthocyanin and flavonol derivatives.

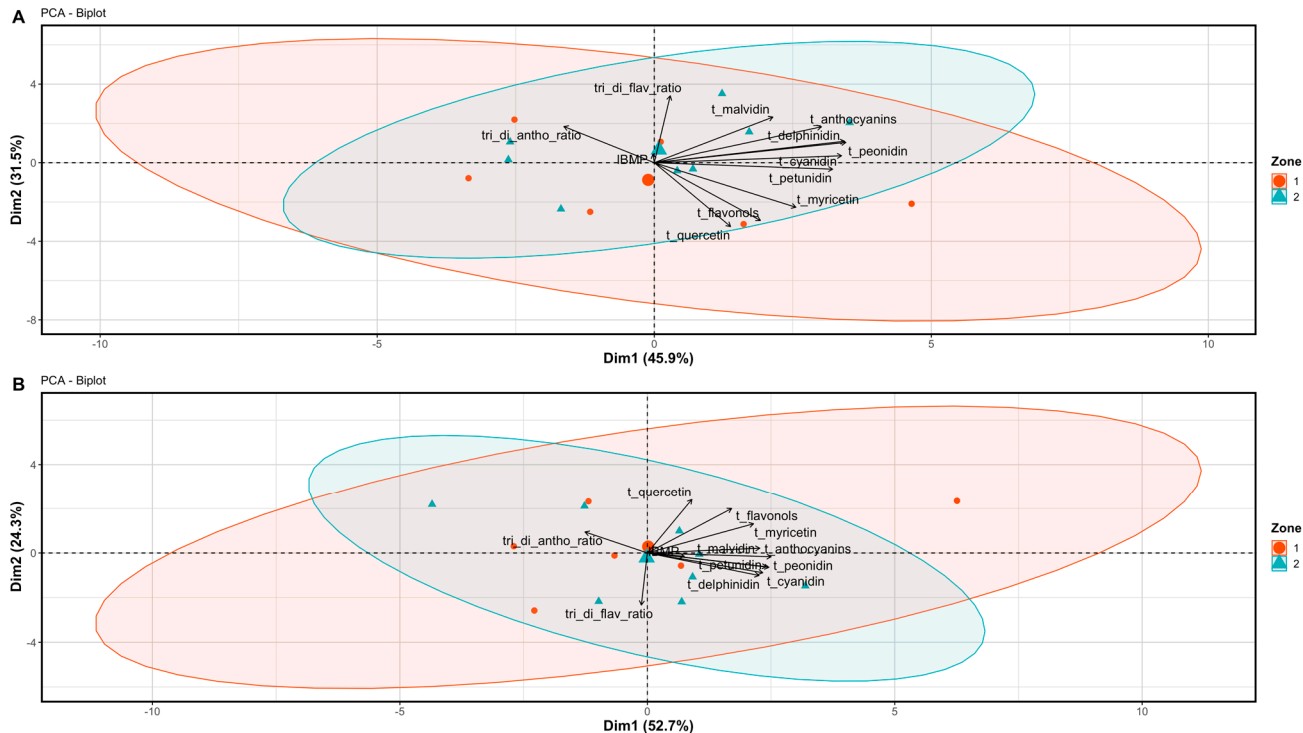

**Figure 2.** Bi-plot from the principal component analysis (PCA) discriminating the two plant water status zones within a commercial vineyard in Oakville, CA, USA, in (**A**) 2019 and (**B**) 2020, according to grape berry skin flavonoid composition at harvest, including t_anthocyanins: total skin anthocyanins; t_flavonols: total skin flavonols; tri_di_antho_ratio: anthocyanin tri- to di-hydroxylation ratio; tri_di_flav_ratio: flavonol tri- to di-hydroxylation ratio flavonol, flavonols, and anthocyanin derivatives (quercetin, myricetin, cyanidin, delphinidin, peonidin, petunidin, malvidin). Point sizes represent the concentrations of corresponding compounds.

### 3.3. Wine Aromatic Profiles

To investigate the effect of zoning on wine aromatic compounds, PCA analyses were performed, and both zones clustered differently in both years (Figure 3). In 2019, the first two principal components (PC) explained 78.9% of the total variation in the dataset, and PC1 and PC2 accounted for 49.3% and 29.6% of the total variation, respectively (Figure 3A).

The difference in Zone 1 was explained by a higher concentration of isobutyric acid, a chain fatty acid, while Zone 2 was characterized by higher amounts of alcohols, including benzyl alcohol, farsenol, geraniol, phenylethyl alcohol, and nerolidol; esters, including ethyl decanoate, ethyl hexanoate, and ethyl octanoate; and terpenes, including α-terpinene, β-damascenone, β-myrcene. In 2020, the first two PCs accounted for 98.9% of the total variation in the dataset, and PC1 and PC2 accounted for 78.1% and 20.8% of the total variation, respectively (Figure 3B). Zone 1 was characterized to have more benzyl alcohol, farnesol, geraniol, hexanol, nerolidol, and octen-3-ol along with terpene and β-myrcene. Zone 2 was characterized to more isoamyl alcohol, isobutanol, linalool, and phenylethyl alcohol; esters, including ethyl decanoate and ethyl octanoate; and the terpene β-damascenone. In both years, the aromatic compounds that were predominant in each zone were significantly and positively correlated with each other, but negatively correlated with the aromatic compounds predominant in the other zone. Except for 2020, benzyl alcohol did not have a significant correlation with either group of aromatic compounds in each zone.

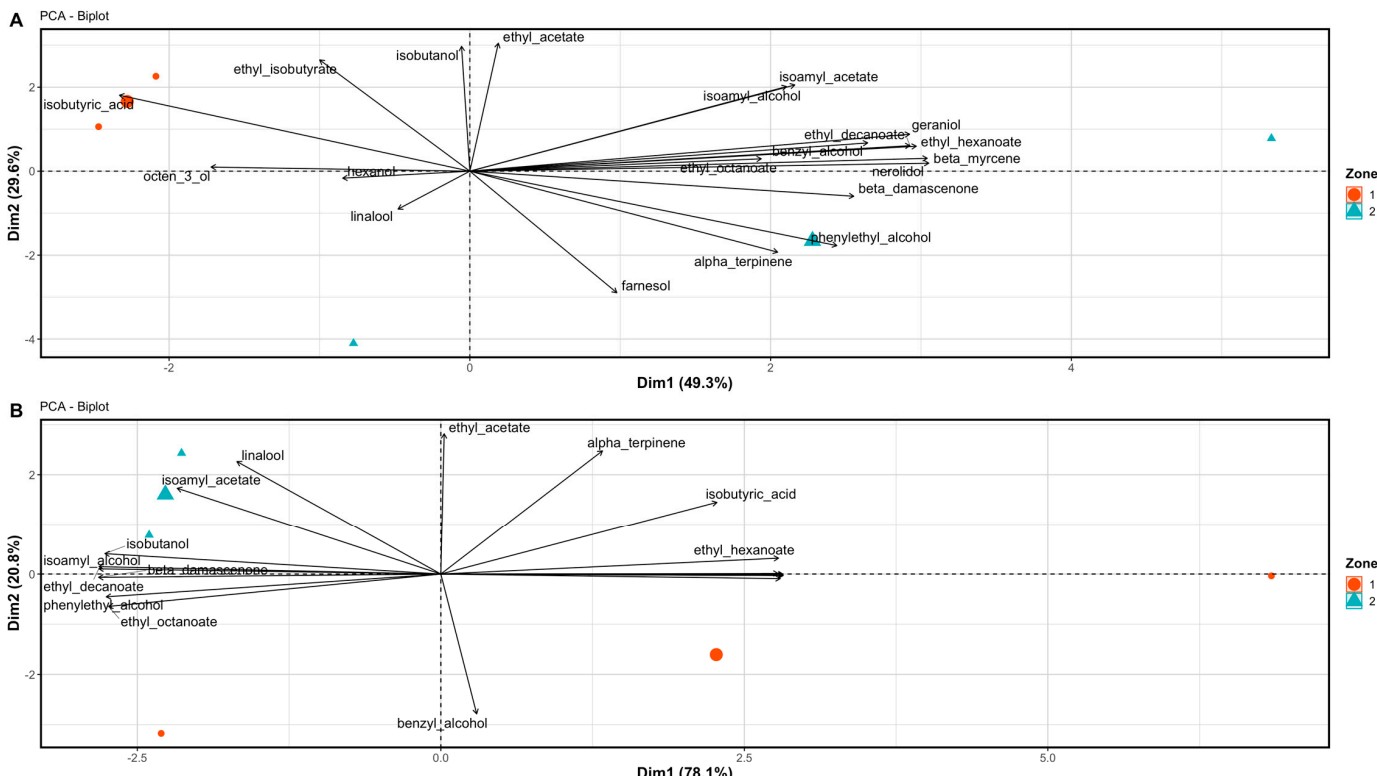

**Figure 3.** Bi-plot from the principal component analysis (PCA) discriminating the two plant water status zones on wine aromatic compounds within a commercial vineyard in Oakville, CA, USA, in (**A**) 2019 and (**B**) 2020, according to alcohol (benzyl alcohol, farnesol, geraniol, hexanol, nerolidol, octen-3-ol, and phenylethyl alcohol), ester (ethyl acetate, ethyl decanoate, ethyl hexanoate, ethyl isobutyrate, ethyl octanoate, and isoamyl acetate), monoterpene (α-terpinene, β-damascenone, β-myrcene, and linalool), and chain fatty acids (isobutyric acid). Point sizes represent the concentrations of corresponding compounds.

## 4. Discussion

### 4.1. The Potential of Differential Harvest in Managing Spatial Variability in Plant Physiology

Vineyard spatial variability in plant physiology and berry/wine chemistry mostly derives from vineyard topography and soil characteristics [21]. Differential harvest has shown its ability to capture the significant differences in berry secondary metabolism [21] as well as in wine flavonoid concentration [18], which could potentially provide opportunities to minimize differences within the vineyard. Also, this approach can offer the ability to alleviate logistic issues regarding allocating and storing fruits [33]. In general, this

approach can provide an instantaneous response regarding differences in composition and bridge the gap in vineyard productivity and wine composition spatially. Although many approaches have been investigated to delineate vineyards to guide differential harvest, including the variables assessed by proximal/remote sensing [34,35], yield [36,37], and soil characteristics [38], mapping grapevine physiological parameters showed great potential due to its direct effects on grapevine physiology and berry chemical composition [39,40].

Among these physiological parameters, plant water status was the one variable that was focused on to delineate vineyards into management zones because of the significance of plant water status in grapevine vegetative growth as well as berry/wine flavonoid and aromatic compound accumulation [41,42]. This approach provided reliable in capturing the spatial variability and the immediate connection in both plant physiology and berry/wine chemistry, as shown in previous studies [21,23]. Brillante et al. (2017) used season-long plant water status to generate two management zones in their experimental vineyard, and the more-water-stressed zone showed significant difference in leaf photosynthetic activities and berry primary metabolism and secondary metabolism [21]. In that study, there was a difference in wine flavonoid concentration as well, which confirmed that the two zones would have uneven berry and wine chemical compositions if harvested at the same time, causing the overall quality to be subpar [43]. There was other evidence of delineating vineyard by plant water status, which mostly resulted in similarly close relationships between plant water status and grapevine development [44]. However, there is one disadvantage of using plant water status to direct vineyard delineation, which is that its on-site measurement would be time-consuming and labor-intensive.

Nevertheless, long-term plant water status can be computed to reveal the spatial distribution in water status variations, and it would still be more easily approachable than mapping the vineyard by yield or berry variables (i.e., berry maturity) due to its compatibility with proximal/remote sensing, as shown in previous studies [40,45]. These studies investigated the relationships between proximal/remote sensing technologies and plant water status spatially, which can be beneficial to assist in and eventually accelerate capturing the spatial variability in plant water status, making it more efficient to derive vineyard management zones for a more instantaneous response in directing harvest. In the previous section of the same study, the season-long plant water status in 2018 did not only show the variability in grapevine physiology, but also can be related to soil electrical conductivity [23]. Therefore, as a plant physiological variable that can be intimately connected to vineyard soil conditions and determine grapevine physiological growth and development at the same time, season-long plant water status can be used to delineate vineyards.

The present study did not show that delineating vineyard based on soil $EC_a$ sensing at harvest can effectively reveal the spatial variability in berry chemistry. However, one previous study noticed that soil $EC_a$ sensing in both deep and shallow soils can be linked to berry total skin anthocyanins [18]. Considering the tight relationships between soil $EC_a$ and soil water content and soil texture, which directly determine grapevine water status [46], the close relationships between soil proximal sensing and berry/wine flavonoid and aromatic compounds are highly probable. However, we need more evidence to verify if soil proximal sensing can be directly used or integrated into vineyard delineation for differential management strategies. Nevertheless, when assessing soil $EC_a$ at the deep-soil layer, its relationship with berry chemistry was relatively noticeable; this phenomenon was confirmed by previous studies [18]. Other studies previously discussed that precipitation before flowering can determine the effectiveness of canopy management strategies [47], which suggests that soil water content might play a significant role in directing grapevine development. This might be the reason that deep-soil $EC_a$ showed more connections with plant physiological parameters than shallow-soil $EC_a$.

On an individual plant scale, plant water status is extremely critical in determining grapevine physiological development, including vegetative growth [42] and berry development [48]. As reported previously, spatial variability in plant water status could still be

linked to the spatial distribution of plant physiological parameters [21]. In our study, Zone 2 showed a higher cluster number per vine and pruning weight, which agreed with previous studies, where higher plant water status contributed to more vegetative growth [42]. Also, higher TA values were measured in Zone 2 in the first season of this study, which agrees with what was observed in previous studies [49,50]. However, there were not many differences observed between the two zones in yield components or berry primary metabolites. One reason could be that the vineyard was delineated based on season-long plant water status in 2018, meaning that the differences in grapevine physiology, derived from the spatial variability, might not be completely translated into the later seasons. This can be confirmed by the leaf gas exchange measurements in both 2019 and 2020 between the two zones, where there was no difference observed in any of the gas exchange parameters except that only $A_n$ was higher in Zone 2 in 2020.

### 4.2. The Potential of Differential Harvest in Managing Spatial Variability in Berry and Wine Chemistry

In this study, there were minimal differences in berry skin anthocyanins and no differences in berry skin flavonols between the two zones. This might be because the differences in plant water status between the two zones were not severe enough to separate them in berry skin flavonoid concentration. Berry skin anthocyanins and flavonols were shown to not be as sensitive when the water deficit was not severe enough to alter their concentrations in berries [51]. Interestingly, the chemical profiles in the wines from the two zones were significantly different in both years, but the two years had different patterns in wine flavonoids. Zone 2 generally showing greater anthocyanin and flavonol concentrations in 2019 but not 2020 could be attributed to the more advanced berry maturity when they were harvested in 2019 compared to 2020. It was also shown that the longer "hang time" of fruits on grapevines advanced the flavonoid degradation when the TSS exceeded approximately 25 °Brix [52], which might have contributed to the discrepancy between the two seasons in this study. It was shown that the plant water status or berry maturity directed by plant water status could also alter the berry skin porosity [53], berry mass or size [42], or skin-to-berry ratio [54], leading to a different extractability of flavonoids into the wine, which could explain why there was no significant difference observed in the berry skins but there was a difference in the resultant wine systems, as well as the discrepant differences observed between the two seasons.

Some of the aromatic compounds were altered by differences in plant water status in this study. Ou et al. (2010) reported that water deficits decreased β-damascenone, which was observed in our study as well [13]. In another works, terpenes and alcohols were greater in number with lower plant water status [55], which was partially corroborated in our study. Although the families of the wine aromatic compounds were not distinctively different, all the compounds still grouped distinctively within each zone between the two water status zones, indicating that the wine aromatic profile of one zone was significantly different from that of the other one. Most of the terpenes and esters accumulated in the less-water-stressed zone, suggesting that higher water stress might diminish these flavors in wine. Contrarily, previous research reported that water deficit increased esters and terpenes in Syrah, Merlot, Tocai Friulano, and Cabernet Sauvignon wines, although this result was dependent on the growing season [14,15,56]. This can be attributed to the specific water deficit levels in the grapevines from these studies. As the California climate was hot and dry in 2019 and 2020, the biodegradation of these aromatic compounds might have been promoted, which caused the discrepancy in the relationships observed in this study, and that between these compounds and water deficits in the previous study.

## 5. Conclusions

In this study, we confirmed that, by delineating the vineyard into two management zones based on season-long plant water status, we can the spatial variability in plant physiology and berry/wine chemical profiles. Although the difference in plant physiology

and berry chemistry was not constituently significant, the differences in wine flavonoid concentration and aromatic profiles were significant between the two zones. This provided evidence of vineyard delineation based on plant water status being effective in revealing the spatial variability in wine flavonoid and aromatic compounds, which can provide directions to growers to utilize such a method as a viable approach to minimize spatial variability in grapevine production and wine chemistry. This approach can also offer opportunities for large-acreage wine-grape growers to achieve a higher level of homogeneity in wine flavonoid concentration and aromatic profiles in vineyards.

**Author Contributions:** Conceptualization, S.K.K. and R.Y.; methodology, S.K.K. and R.Y.; software, R.Y. and N.T.; validation, S.K.K., N.T. and R.Y.; formal analysis, R.Y.; investigation, S.K.K., N.T. and R.Y.; resources, S.K.K.; data curation, R.Y.; writing—original draft preparation, R.Y.; writing—review and editing, S.K.K., N.T. and R.Y.; visualization, S.K.K. and R.Y.; supervision, S.K.K.; project administration, S.K.K. and R.Y.; funding acquisition, S.K.K. All authors have read and agreed to the published version of the manuscript.

**Funding:** This research received no external funding.

**Data Availability Statement:** The raw data supporting the conclusions of this article are available on request to the corresponding author.

**Conflicts of Interest:** Author Sahap Kaan Kurtural was employed by the company Kurtural Vineyard Consulting. The remaining authors declare that the research was conducted in the absence of any commercial or financial relationships that could be construed as a potential conflict of interest.

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
