# Peer review of "Obtaining Spatial Variations in Cabernet Sauvignon (Vitis vinifera L.) Wine Flavonoid Composition and Aromatic Profiles by Studying Long-Term Plant Water Status in Hyper-Arid Seasons"

_horticulturae, doi:10.3390/horticulturae10010068_

Round 1

Reviewer 1 Report

Comments and Suggestions for Authors

Spatial variation is an interesting topic. The vineyard site is perfect for the study. But the manuscript should be further polished.

Line 58, if there are spatial variation, it is common that some vineyards will be managed in different ways, such as different irrigation, or row orientation. 

Line 84, give the latitude and longitude of the site.

Line 89, canopy management involves a range of techniques used in the vineyard to manage a grapevine’s canopy (the leaves, shoots and fruit) from the time of winter pruning until harvest. It's impossible to have no further canopy management in 2019 and 2020. 

Line 107, 30 m*30 m contained 14 experimental units. Each unit will be 8*8m. The vine * row distance is 1.5*2m. Therefore, there should be 21 vines in each experimental unit. I don't understand why 3 individual adjacent grapevines in each experimental unit.

Line 147, it's not reasonable to harvest on the same day in both years. The Brix of 2020 was quite lower than 2019. 

Line 214, 'Chemicals' may not need to be described separately. 

Line 335 & 365, the dots size of the same zone are different. It should be explained in footnotes. 

No the section of discussion. 

Author Response

Spatial variation is an interesting topic. The vineyard site is perfect for the study. But the manuscript should be further polished.

Line 58, if there are spatial variation, it is common that some vineyards will be managed in different ways, such as different irrigation, or row orientation. 

Response: We appreciate this comment from the reviewer. For this experimental block used in this study specifically, it was only 1-acre (0.4 hectare) vineyard block that was managed uniformly, including irrigation, row orientation, etc. The spatial variability was solely derived from soil at the experimental site.

Line 84, give the latitude and longitude of the site.

Response: We have added the latitude and the longitude of the site to the manuscript.

Line 89, canopy management involves a range of techniques used in the vineyard to manage a grapevine’s canopy (the leaves, shoots and fruit) from the time of winter pruning until harvest. It's impossible to have no further canopy management in 2019 and 2020. 

Response: We appreciate this comment from the reviewer. However, there was no further canopy management strategies after shoot thinning in both years. We tried to keep sufficient leaf coverage to avoid sunburns on those berries. We did not perform cluster thinning either because we were trying to get decent production from that block. On top of that, the irrigation at 50% ETc and the warmer climate kept the canopy development in check, and with the single high wire system, the shoots were floppy and did not require hedging either.

Line 107, 30 m*30 m contained 14 experimental units. Each unit will be 8*8m. The vine * row distance is 1.5*2m. Therefore, there should be 21 vines in each experimental unit. I don't understand why 3 individual adjacent grapevines in each experimental unit.

Response: We appreciate this comment from the reviewer and the reviewer’s attentiveness. However, there was a misunderstanding here. The fish-net grid was on 30*30m, which means that each experimental unit was 30 m from the closest ones. The fish-net grid was generated in GPS software and overlaid with our vineyard block map, that was how we labeled the 14 experimental units. However, we still wanted to have some replicates within each experimental unit, that was why we included 3 vines within each one. Hence, 14 experimental units with 3 vines in each, 42 vines in total were used for the on-site measurements.

Line 147, it's not reasonable to harvest on the same day in both years. The Brix of 2020 was quite lower than 2019. 

Response: We appreciate this comment from the reviewer. For the vineyard block, the fruits were grown for commercial uses, we had to consider logistics from the University of California as well as the weather to avoid rainfalls in September. All the reasons made us harvest the fruits around the same time, at the end of September in each season.

Line 214, 'Chemicals' may not need to be described separately. 

Response: We think it would be better to leave this section here separately, so with the sufficient information, all the chemical analyses can be repeatable. 

Line 335 & 365, the dots size of the same zone are different. It should be explained in footnotes. 

Response: We have added brief explanations in the captions of the two figures.

No the section of discussion. 

Response: We apologize for missing this section. The discussion and conclusion sections were accidently lost when submitting to the journal. We put the two big sections back to this manuscript with our updated reference list.

We have marked the modifications in color red and updated the reference list, please do not hesitate to contact us if you have any more suggestions to improve this manuscript. Again, we truly appreciate the reviews time and efforts spent on reviewing our manuscript.

Reviewer 2 Report

Comments and Suggestions for Authors

The article presented for review raises very important issues related to cultivation in conditions of water deficit and its impact on the potential of the flavonoid composition and the aromatic profile of Cabernet Sauvignon wine. Weather conditions, including mainly very different rainfall totals in individual years of research, made it possible to thoroughly investigate the problem. I have no objections to the research methodology and analyzes performed, they are consistent with this type of scientific work. As an agrotechnician, the above work lacks data regarding the yield and the number of potential wine settings - they are crucial to illustrate production possibilities and efficiency. If it is possible at this stage, please complete these parameters.

The work is very interesting and may be approved for publication after minor changes.

Author Response

We truly appreciate the reviews time and efforts spent on reviewing our manuscript.

At this stage, we cannot perform additional analyses on the wine samples, and the yield data was previously published so we did not want to use it here. The currently included data in this manuscript are sufficient in our opinions to express what we are trying to convey to the readers, including berry and wine flavonoid and aromatic concentration and composition. We have marked the modifications in color red and updated the reference list, please do not hesitate to contact us if you have any more suggestions to improve this manuscript. 

Reviewer 3 Report

Comments and Suggestions for Authors

The authors study the effect of variability of vineyard soil in the water stress on the Cabernet Sauvignon variety and how it affects part of the chemical composition of the grapes and wines obtained. Authors observed significant differences  between the two areas studied  The work is well presented, the data are correctly discussed, with a statistical treatment that supports this discussion. 

Author Response

We truly appreciate the reviews time and efforts spent on reviewing our manuscript. We have marked the modifications in color red and updated the reference list, please do not hesitate to contact us if you have any more suggestions to improve this manuscript.